# Global and Sex-Stratified Genome-Wide Association Study of Long COVID Based on Patient-Driven Symptom Recall

**DOI:** 10.3390/ijms26189252

**Published:** 2025-09-22

**Authors:** Sara Polo-Alonso, Álvaro Hernáez, Irene R. Dégano, Ruth Martí-Lluch, Mel·lina Pinsach-Abuin, Roberto Elosua, Isaac Subirana, Marta Puigmulé, Alexandra Pérez, Raquel Cruz, Silvia Diz-de Almeida, Eulàlia Puigdecant, Elisabet Selga, Xavier Nogues, Joan Ramon Masclans, Roberto Güerri-Fernández, Héctor Cubero-Gallego, Helena Tizon-Marcos, Beatriz Vaquerizo, Ramon Brugada, Rafel Ramos, Anna Camps-Vilaró, Jaume Marrugat

**Affiliations:** 1Registre Gironí del Cor (REGICOR) Study Group, Hospital del Mar Research Institute, 08003 Barcelona, Spain; 2PhD Program in Biomedicine, Universitat Pompeu Fabra, 08003 Barcelona, Spain; 3Centro de Investigación Biomédica en Red de Enfermedades Cardiovasculares (CIBERCV), Instituto de Salud Carlos III, 28029 Madrid, Spain; 4Blanquerna School of Health Sciences, University Ramon Llull, 08022 Barcelona, Spain; 5Faculty of Medicine, University of Vic-Central University of Catalonia, 08500 Vic, Spain; 6Institute for Research and Innovation in Life Sciences and Health in Central Catalonia (IRIS-CC), 08500 Vic, Spain; 7Vascular Health Research Group, Institut Universitari per a la Recerca en Atenció Primària Jordi Gol i Gurina, 17002 Girona, Spain; 8Girona Biomedical Research Institute, 17190 Girona, Spain; 9Cardiovascular Genetics Center, Institut d’Investigació Biomèdica de Girona Dr. Josep Trueta (IdIBGi), 17190 Salt, Spain; 10Cardiovascular Epidemiology and Genetics Group, Hospital del Mar Research Institute, 08003 Barcelona, Spain; 11Centro Singular de Investigación en Medicina Molecular y Enfermedades Crónicas (CIMUS), Universidade de Santiago de Compostela, 15782 Santiago de Compostela, Spain; 12Centro de Investigación Biomédica en Red de Enfermedades Raras, Instituto de Salud Carlos III, 28029 Madrid, Spain; 13Musculoskeletal Research Unit, Hospital del Mar Research Institute, 08003 Barcelona, Spain; 14Department of Internal Medicine, Hospital del Mar, 08003 Barcelona, Spain; 15Centro de Investigación Biomédica en Red de Fragilidad y Envejecimiento Saludable, Instituto de Salud Carlos III, 28029 Madrid, Spain; 16Faculty of Medicine, Universitat Autònoma de Barcelona (UAB), 08193 Bellaterra, Spain; 17Critical Illness Research Group (GREPAC), Hospital del Mar Research Institute, 08003 Barcelona, Spain; 18Department of Critical Care, Hospital del Mar, 08003 Barcelona, Spain; 19Faculty of Medicine and Life Sciences, Universitat Pompeu Fabra (UPF), 08003 Barcelona, Spain; 20Centro de Investigación Biomédica en Red de Enfermedades Infecciosas, Instituto de Salud Carlos III, 28029 Madrid, Spain; 21Department of Infectious Diseases, Hospital del Mar Research Institute, 08003 Barcelona, Spain; 22Biomedical Research in Heart Diseases Group, Hospital del Mar Research Institute, 08003 Barcelona, Spain; 23Department of Cardiology, Hospital del Mar, 08003 Barcelona, Spain; 24Department of Cardiology, Hospital Josep Trueta, University of Girona, 17007 Girona, Spain; 25Department of Medical Science, School of Medicine, University of Girona, 17071 Girona, Spain; 26Primary Care Services, Catalan Institute of Health, 08007 Barcelona, Spain

**Keywords:** post-acute COVID-19 syndrome, COVID-19, SARS-CoV-2, genome-wide association study, genetic polymorphism, sex characteristics

## Abstract

We aimed to explore the global and sex-specific genetic variants associated with long COVID, as defined by patient-driven symptom recall. A 1-year cohort study of 2411 COVID-19 patients collected long COVID symptoms with an open-ended, non-directed questionnaire, and long COVID incidence was determined according to the World Health Organization definition. Global and sex-stratified genome-wide association analyses were conducted by logistic regression models adjusted for age, sex (in the global analysis), and the first 10 principal components. We assessed sex-variant interactions and performed gene-based analyses, gene mapping, and gene-set enrichment analyses. When comparing the 1392 long COVID cases with the non-cases, we identified 23 lead variants from suggestive signals: 13 from the global analysis, 5 from females, and 5 from males. Five variants showed a significant interaction with sex (two in females, three in males). We mapped 15 protein-coding genes related to diseases of the immune and nervous systems and tumoral processes. Notably, CD5 and VPS37C, linked to immune function, were significantly associated with long COVID in men. Our results suggest that persistent immune dysregulation may be involved in the development of precisely defined long COVID.

## 1. Introduction

Since the beginning of the COVID-19 pandemic, many patients have developed persistent symptoms lasting months after acute infection, a condition now recognized as long COVID or post-acute sequelae of SARS-CoV-2 infection [1,2]. It is estimated that between 6% and 70% of individuals infected with SARS-CoV-2 develop long COVID [3,4]. It includes a variety of symptoms such as fatigue, dyspnea, chest pain, anxiety, depression, fever, and hair loss, many of which limit quality of life [3,5,6,7,8].

Understanding the biological mechanisms driving severe and long COVID, potentially linked to immune dysregulation and its consequences [9,10], is crucial for developing targeted therapeutic strategies. Genetics could provide valuable insights into these mechanisms [11]. The COVID-19 Host Genetics Initiative recently carried out a trans-ancestry GWAS meta-analysis, applying a strict case definition (test-verified infection) versus broad population controls, and identified a genome-wide significant association at the *FOXP4* locus in 3018 strict cases versus 994,582 controls [12]. However, by aggregating studies with variable definitions of long COVID, follow-up intervals, and limited sex stratification, such meta-analyses cannot uniformly capture the depth or temporal dynamics of long COVID symptoms. Moreover, its checklist-based phenotyping cannot distinguish persistent, patient-reported sequelae from prompted symptom endorsement, potentially diluting the long COVID case definition. Here, we present a prospectively recruited GWAS of long COVID in 2411 RT-PCR- or antigen-confirmed SARS-CoV-2-positive individuals from Catalonia, Spain, applying an open-ended, one-year follow-up questionnaire to capture 24 persistent symptoms with a homogeneous clinical definition. This open-ended, clinician-administered interview allows unaided symptom recall, thereby yielding a deeply phenotyped outcome less prone to misclassification. We aimed to analyze the genetic variants associated with the development of long COVID in female and male separately, as well as in both sexes combined, and to deepen the understanding of the biological mechanisms underlying the condition.

## 2. Results

### 2.1. Participants

2411 individuals were eligible for further analysis, 1392 suffered from long COVID, and 1019 did not (Figure 1). As seen in Table 1, individuals with long COVID were younger than those without long COVID. Females with long COVID showed a lower prevalence of hypertension, while males with long COVID experienced a more severe acute phase of the infection and showed a different ancestry distribution.

### 2.2. Genome-Wide Association Analysis and Variant–Sex Interaction

Several suggestive signals (*p*-value < 5 × 10^−6^) were identified (Figure 2). In the global analysis, 13 genomic loci were located on chromosomes 1, 2, 3, 4, 8, and 12. Sex-stratified analyses revealed 5 genomic loci in females (chromosomes 4, 9, 17, and 19) and in males (chromosomes 1, 4, 11, 19, and 21). 23 lead genetic variants were identified, (13 in the global analysis, 5 in females, and 5 in males; Table 2), and none were shared across the analyses. The gene variant rs10888603, identified in males, exhibited the lowest *p*-value (5.2 × 10^−8^). In the sensitivity analysis results (Appendix A), the three genomic risk loci described in males on chromosomes 4, 11, and 21 (Appendix A) were confirmed. For females, no clear genetic signals were found.

A statistically significant interaction with sex was observed in five genetic variants: two in females (rs146309770 and rs915401) and 3 in males (rs1274686, rs2186409, and rs2717199). The significance *p*-value threshold was established at a Bonferroni-adjusted *p*-value < 0.05.

### 2.3. Functional Annotation

Positional and expression quantitative trait locus (eQTL) gene-mapping of protein-coding genes was performed within the genomic regions defined by the five lead genetic variants that showed a statistically significant interaction with sex (Table 2). In females, rs915401 mapped to the gene Laminin Subunit Gamma 3 (*LAMC3*), and rs146309770 did not map to any protein-coding gene. In males, rs2186409 mapped to seven protein-coding genes: five by eQTL (Transmembrane Protein 109 (*TMEM109*), Pepsinogen A3 (*PGA3*), Pepsinogen A5 (*PGA5*), Von Willebrand Factor C and EGF Domains (*VWCE*), and Transmembrane Protein 216 (*TMEM216*)), and two by both position and eQTL (Vacuolar Protein Sorting-Associated Protein 37C (*VPS37C*) and *CD5*). rs1274686 mapped to seven protein-coding genes: five by position (Myosin Binding Protein C2 (*MYBPC2*), ER Membrane Protein Complex Subunit 10 (*EMC10*), Josephin Domain Containing 2 (*JOSD2*), Aspartate Dehydrogenase Domain Containing (*ASPDH*), and Leucine Rich Repeat Containing 4B (*LRRC4B*)), one by eQTL (Prostate Tumor Overexpressed 1 (*PTOV1*)), and one by both position and eQTL (Family with sequence similarity 71 member E1 (*FAM71E1*)). rs2717199 was not mapped to any protein-coding gene.

Then, the gene set enrichment analysis was performed with the protein-mapped genes. In the case of females, it could not be assessed because only one gene was mapped and a minimum of two were needed for the analysis. In males, there was an overrepresentation of the target genes of the transcription factors protein inhibitor of activated STAT 4 (*FAM71E1*, *EMC10*, and *JOSD2*), zinc finger protein 524 (*VPS37C*, *PTOV1*, *FAM71E1*, and *EMC10*), and the gene set Nikolsky breast cancer 11q12-q14 amplicon (*TMEM109*, *CD5*, *VPS37C*, *PGA5*, *VWCE*, and *TMEM216*).

Finally, the gene-based analysis performed with MAGMA revealed two genes that were significantly associated with long COVID in males (Appendix A): *CD5* (*p*-value 2.16 × 10^−7^) and *VPS37C* (*p*-value 2.97 × 10^−7^). The gene-based analysis in females did not find any gene with statistical significance.

## 3. Discussion

In this study, by incorporating an open-ended long COVID symptoms questionnaire, we obtained highly accurate phenotypic data and performed sex-stratified analyses. Although no genetic variants reached genome-wide significant level, suggestive signals (*p*-value < 5 × 10^−6^) were found in the three GWAS, supporting the idea that genetic susceptibility to long COVID may be sex-specific.

Our gene-mapping analyses identified 15 genes associated with these suggestive variants (1 in females: *LAMC3*; and 14 in males). Of particular interest, *CD5* and *VPS37C*, were not only identified through positional and eQTL mapping, but also reached statistical significance in the MAGMA gene-based analysis. These genes are linked to immune system function. *CD5* encodes the CD5 protein, expressed in T cells, B1a cells, chronic lymphocytic leukemia cells, and dendritic cells [13]. Some studies have reported the development of chronic lymphocytic leukemia following SARS-CoV-2 infection [14,15,16]. A possible explanation is that mutant chronic lymphocytic leukemia cells may exist as small niches in healthy individuals and expand following SARS-CoV-2–induced cytokine dysregulation [15]. Similarly, *VPS37C* is included in the transcriptomic signature that differentiates multisystem inflammatory syndrome in children (occurring weeks after SARS-CoV-2 infection) from Kawasaki disease and other infections [17]. The involvement of *CD5* and *VPS37C* suggests that long COVID may be influenced by sustained immune dysregulation after the initial infection. Other mapped genes also support an immune-related mechanism. For instance, *FAM71E1* and *EMC10* have been associated with lupus erythematosus [18] and genes like *PGA3*, *VWCE*, and *PTOV1* are associated with immune infiltrates and immunity in various tumors [19,20,21]. Taken together, these findings suggest that persistent immune system alterations, possibly triggered by SARS-CoV-2, may play a critical role in the development of long COVID. This aligns with previous studies showing that interleukin-6 has been proposed as a predictive biomarker of long COVID risk [22]. Some of the identified genes may be involved in neurological aspects of long COVID. Variants in *EMC10* have been related to neurodevelopmental disorders and intellectual disability [23], and alterations in *LAMC3* have been linked to cortical development anomalies and epilepsy [24]. An association between central nervous system-related genes and long COVID may be compatible due to the frequent neurological symptoms seen in the disease [25], and neurocentric proteomic and microglial studies further support a link between viral persistence and long-term cognitive sequelae [26,27].

In addition, several genes mapped in this study are related to tumor progression and cancer prognosis. *JOSD2* influences the proliferation and progression of hepatocarcinoma, lung cancer, and esophageal squamous cell carcinoma [28,29,30] and modulates acute myeloid leukemia progression [31]. *PGA3* is highly expressed in bone metastases of gastric cancer, indicating poor survival [32], and *PGA5* expression can induce changes related to tumor progression and epithelial–mesenchymal transition [33]. *VWCE* belongs to a 3-gene model proposed as a survival signature of uterine cancers [34]. *MYBPC2*, *TMEM109*, and *LACM3* have also been associated with various tumor types [35,36,37]. While the link between long COVID and cancer risk or progression remains speculative, these findings highlight pathways worth exploring further.

Additionally, in contrast to the recent meta-analysis by Lammi et al. [12], which reported a genome-wide significant association at the FOXP4 locus, we did not observe this signal in our study. A key methodological difference is that Lammi et al. compared long COVID cases with controls drawn from the general population, which included both SARS-CoV-2–infected and non-infected individuals. Interestingly, when they restricted the analysis to controls with confirmed SARS-CoV-2 infection, the FOXP4 association did not reach statistical significance. Our study design, which compared long COVID cases exclusively against SARS-CoV-2–infected non-cases, follows this latter approach and may explain the absence of association at the FOXP4 locus in our results. This highlights how the choice of control group can influence the detection of genetic associations in long COVID research.

The main strengths of our study are, first, the use of an open-ended, clinician-administered interview that lets participants report symptoms freely, yielding a precise, checklist-free definition of long COVID; and second, our sex-stratified GWAS and gene variant-by-sex interaction analyses, which uncover genetic signals that differ between female and male. However, our study also has limitations. Firstly, long COVID characterization is recent and still evolving, encompassing a broad range of symptoms. This heterogeneity complicates the identification of specific genetic associations, as variability in symptom presentation reduces the strength of potential signals. Secondly, our limited sample size may reduce the statistical power to detect genetic signals involved in long COVID. Third, we were unable to conduct a meta-analysis with other available datasets due to differences in phenotype definitions. Fourth, differences in linkage disequilibrium patterns, allele frequencies, and individuals with and without long COVID distributions across ancestries posed additional challenges. Nevertheless, these differences were addressed by including the first 10 principal components as covariates in the association analysis. Fifth, SARS-CoV-2 reinfections were not recorded during follow-up. Reinfection events may exacerbate or re-trigger symptoms, potentially mimicking or amplifying the long COVID phenotype. Future studies should incorporate reinfection surveillance to better disentangle their role in the persistence of symptoms. Finally, our findings may not fully generalize to other populations. All participants were recruited in Catalonia, northeastern Spain, and were 35 to 84 years old, within a universal healthcare system. These factors differ from other settings and could influence how long COVID develops and persists.

## 4. Materials and Methods

### 4.1. Study Design and Participants

The GINA-COVID project is a prospective cohort study that comprised 3073 COVID-19 patients aged 35 to 84 years old. They were recruited from four centers in Catalonia, northeastern Spain [38]. All the participants had tested positive for SARS-CoV-2 using reverse transcription-polymerase chain reaction, rapid antigen, or IgG tests between February 2020 and December 2021. Participants were excluded if they had been vaccinated against SARS-CoV-2 before the diagnosis, if clinical data were not available in the electronic medical record of the Catalan health system (universal coverage), or if they did not participate in the 1-year follow-up for long COVID symptoms.

### 4.2. Definition of Long COVID

We determined long COVID status in accordance with World Health Organization criteria [39]. Trained personnel conducted an open-ended questionnaire on participants’ COVID-19 symptoms one year after diagnosis or one year after discharge from hospital for hospitalized patients. Participants were asked to describe spontaneously any symptoms they had experienced weeks or months after their infection. Trained personnel recorded the presence of symptoms with a predefined list of 24 symptoms previously reported in the bibliography for long COVID (>10% prevalence) [3,7,8,40]. Symptoms reported by participants that were not present in the predefined list were also recorded. For each symptom, participants were asked whether it was present before, during, or after their infection (less than 3 months, up to 3, 6, or 9 months, or up to 12 months or longer). A symptom was classified as persistent if it was absent before the infection but emerged during or after the infection and lasted for at least three months. Symptoms present prior to the infection but worsened during or after the infection for at least three months were also classified as persistent. Following the World Health Organization definition [39], long COVID included individuals who suffered from at least 1 persistent symptom from the 24 listed symptoms.

### 4.3. DNA Collection

DNA was obtained from both peripheral blood and saliva. Peripheral blood was obtained and placed in 4 mL EDTA Anti-Coagulant BD Vacutainer tubes. Saliva samples were collected using the DANASALIVA^®^ Sample Collection kit (DANAGEN^®^, Badalona, Spain). DNA was extracted with either a ChemagicTM^®^ DNA Blood 7k Kit H12 (PerkinElmer^®^, Barcelona, Spain) on a Chemagic MSM I instrument, or a FlexiGene^®^ DNA Kit (Qiagen^®^, Barcelona, Spain) [38].

### 4.4. Genotyping

DNA samples were genotyped with the Axiom Spain Biobank 1 Array (Thermo Fisher Scientific, Waltham, MA, USA), which contains 757,836 genetic variants, including rare and specific variants from the Spanish population. The genotyping process was performed at the Santiago de Compostela Node of the National Genotyping Center (CeGen-ISCIII; https://www.xenomica.eu/servicios/centro-nacional-de-genotipado/, accessed on 14 January 2025) following the manufacturer’s instructions. Genotyping calling and clustering were performed with Axiom Analysis Suite v5.3.0.45.

### 4.5. Quality Control of Genotype Data

Post-genotyping quality control was performed with R.4.3.2 [41] and PLINK v1.9 and v2.0 [42]. Variants with minor allele frequency < 1%, call rate < 98%, or those deviating from Hardy–Weinberg equilibrium (*p*-value < 10^−6^) were excluded. For samples, those with call rate < 98% and a heterozygosity rate deviating by more than five standard deviations from the mean heterozygosity rate of the study were removed. Kinship and ancestry were assessed by pruning autosomal variants with minor allele frequency > 5% (window size of 1000 markers, step size of 80, and r2 of 0.1), obtaining a subset of 72,546 genetic variants. High linkage disequilibrium regions previously described by Price et al. [43] were also removed. We performed identity-by-descent analysis with this set of variants and removed one individual, the one with the highest missingness rate, from each pair with first- and second-degree of kinship (PI_HAT > 0.25). Principal component analysis was conducted using the genetic variants from unrelated individuals to study population stratification and identify outliers for exclusion. Ancestry was analyzed by Admixture [44] on unrelated individuals using the 1000 Genomes Project data [45]. Participants were assigned to a specific ancestry group if their probability of belonging to that group was ≥80%; otherwise, they were categorized as “mixed”. Overall, 2411 individuals (1392 with long COVID and 1019 without long COVID) and 584,906 genetic variants passed quality control.

### 4.6. Variant Imputation

Genetic variants were imputed using the TOPMed version r3 reference panel (GRCh38) through the TOPMed Imputation Server [46]. Post-imputation filtering was applied, retaining variants with an INFO sCORE > 0.8 and a minor allele frequency > 1%, resulting in 8,681,812 variants for further analysis.

### 4.7. Clinical Data

During admission or at the outpatient clinic, age, sex, weight, height, self-reported history of tobacco use (whether the participant was a current smoker or not), and history of diabetes, hypertension, and dyslipidemia were collected from electronic medical records. Body mass index was calculated by dividing an individual’s weight (in kilograms) by the square of their height (in meters) with the following categories: underweight (<18.5 kg/m^2^), normal weight (18.5 to <25.0 kg/m^2^), overweight (25.0 to <30.0 kg/m^2^), and obesity (≥30.0 kg/m^2^).

### 4.8. Statistical Analyses

Patient characteristics were presented as frequencies for categorical variables, while continuous variables were summarized as means with standard deviations for normally distributed data. Comparisons between people with and without long COVID were performed using *t*-tests for means and the chi-square test for categorical variables.

Global and sex-stratified genome-wide association tests were computed by fitting a logistic regression model in Plink 1.9, including age and sex in the global analysis, and the first 10 principal components as covariates. A sex-stratified sensitivity analysis was also performed in European population, including age and the first 10 European-specific principal components as covariates. The genetic variant association significance threshold was set at *p*-value < 5 × 10^−8^.

To formally evaluate effect modification by sex, we analyzed potential interactions between genetic variants and sex in relation to long COVID. To perform this analysis, we selected the lead variants from the global and the sex-stratified GWAS analyses. They were defined as a subset of significant independent variants (*p*-value < 5 × 10^−6^ and r^2^ < 0.6) that were independent at each other at r^2^ < 0.1. Once identified, a logistic regression model was fitted for each of them, in which the age, sex, the first 10 principal components, and the interaction term genetic variant*sex were included as covariates. Bonferroni correction of the *p*-values was performed for multiple testing, and the association significance threshold was established at Bonferroni-adjusted *p*-value < 0.05.

### 4.9. Functional Annotation

The functional annotation, gene mapping, and functional gene-set enrichment analyses were performed with the lead variants showing a significant interaction with sex, in the global or sex-stratified analyses, using SNP2GENE implemented in FUMA (v.1.5.2) [47].

Genomic risk loci included genetic variants (*p*-value < 0.05) in linkage disequilibrium (r^2^ < 0.6) with the independent significant variants, referred to as candidate variants. The maximum distance between linkage disequilibrium blocks to be merged into a genomic locus was 250 kb. The 1000 Genomes Project, Phase 3, from all ancestries [45] was set as the reference panel population, as our sample was multi-ethnic.

Functional consequences of candidate variants on gene function were annotated with Annotate Variation [48], including the combined annotation dependent depletion score for variant deleteriousness [49], Regulome database for potential regulatory functions [50], and gene expression effects were assessed using eQTL of genotype-tissue expression v.8 [51].

To further investigate the biological function of the candidate variants, they were mapped to protein-coding genes and prioritized by two procedures: positional mapping, in which variants were assigned to genes by physical distance (10 kb window), and eQTL mapping, in which genetic variants were mapped to a gene if they had significant effects on its expression, using the genotype-tissue expression v.8 database [51]. Significant variant–gene pairs were established at a false discovery rate < 0.05. Candidate genes were considered for the functional enrichment analysis.

An enrichment analysis of the candidate genes in pre-defined pathways was performed with the GENE2FUNC tool implemented in FUMA. The hypergeometric test was used to test for the overrepresentation of the candidate genes in any of the gene sets, which included Molecular Signatures Database [52,53], WikiPathways [54], and GWAS Catalog [55] information. For a gene set to be overrepresented in the candidate genes, a minimum of two overlapping genes and a Benjamini–Hochberg adjusted *p*-value < 0.05 were needed.

To test for the joint effect of genetic variants mapped to a protein-coding gene, sex-stratified gene-based analyses were performed with MAGMA [56], implemented in FUMA, under the gene variant–wide model with the 1000 Genomes Project, Phase 3 [45], from all the ancestries as the reference panel. For both females and males, genetic variants were mapped to a total of 19,394 protein-coding genes, and the gene-level significance *p*-value threshold was calculated using the Bonferroni method (<0.05/19,394 = 2.578 × 10^−6^).

## 5. Conclusions

Our checklist-free, open-ended phenotyping and sex-stratified GWAS reveal immune-regulatory loci (notably CD5 and VPS37C in men) that point to persistent immune dysregulation as a driver of long COVID. By improving case definition and exposing sex-specific risk, our study helps public health efforts to identify individuals most likely to develop long-term sequelae of SARS-CoV-2 infection. Future work should replicate these signals in larger, harmonized cohorts (ideally applying similarly checklist-free phenotyping and mandatory sex stratification), ultimately guiding research for developing personalized strategies to prevent persistent immune dysregulation after COVID-19 infection.

## Figures and Tables

**Figure 1 ijms-26-09252-f001:**
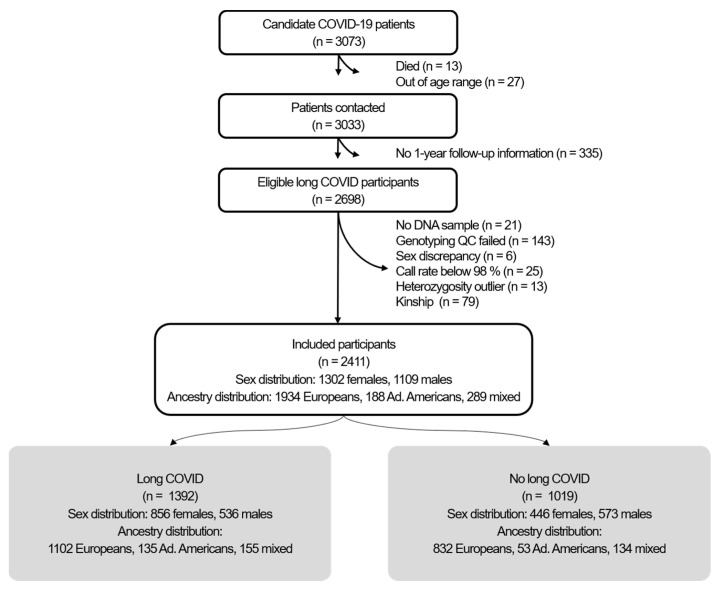
Flow chart of the GINA-COVID study.

**Figure 2 ijms-26-09252-f002:**
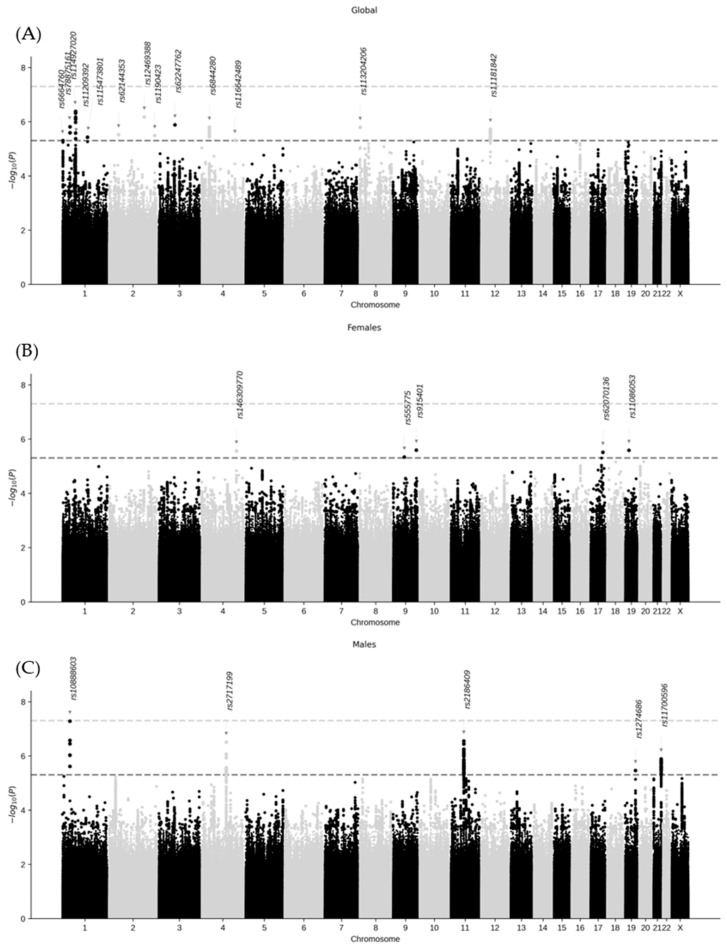
Manhattan plots of long COVID for the global (**A**) and sex-stratified ((**B**) females; (**C**) males) analyses. Lead variants for each association are indicated. Dark gray and light gray lines represent the *p*-value significance thresholds of 5 × 10^−6^ and 5 × 10^−8^, respectively.

**Table 1 ijms-26-09252-t001:** Participants with and without incident long COVID, stratified by sex.

	Females (*n* = 1302)			Males (*n* = 1109)		
	With Long COVID (*n* = 856)	Without Long COVID (*n* = 446)	*p*-Value	With Long COVID (*n* = 536)	Without Long COVID (*n* = 573)	*p*-Value
Age (mean ± SD)	56.4 ± 12.0	58.3 ± 12.6	0.009	57.8 ± 12.2	60.1 ± 12.7	0.002
Severe (hospitalized) acute phase (*n*, %)	196 (23.4%)	91 (20.5%)	0.261	242 (47.3%)	164 (29.6%)	<0.001
Current smokers (*n*, %)	57 (6.70%)	43 (9.71%)	0.070	49 (9.23%)	72 (12.7%)	0.086
Diabetes mellitus (*n*, %)	86 (10.0%)	44 (9.87%)	0.995	80 (14.9%)	99 (17.3%)	0.320
Hypertension (*n*, %)	202 (23.6%)	144 (32.3%)	0.001	209 (39.0%)	225 (39.3%)	0.956
Dyslipidemia (*n*, %)	201 (23.5%)	121 (27.1%)	0.175	179 (33.4%)	194 (34.0%)	0.888
BMI (categorized):			0.534			0.234
<18.5 kg/m^2^ (*n*, %)	6 (0.72%)	6 (1.40%)		2 (0.38%)	1 (0.18%)	
18.5–24.9 kg/m^2^ (*n*, %)	286 (34.3%)	155 (36.1%)		109 (20.7%)	137 (25.0%)	
25.0–29.9 kg/m^2^ (*n*, %)	268 (32.1%)	127 (29.6%)		266 (50.5%)	277 (50.5%)	
≥30.0 kg/m^2^ (*n*, %)	275 (32.9%)	141 (32.9%)		150 (28.5%)	133 (24.3%)	
Ancestry:			0.112			0.002
Admixed Americans (*n*, %)	89 (10.4%)	31 (6.95%)		46 (8.58%)	22 (3.84%)	
Europeans (*n*, %)	677 (79.1%)	370 (83.0%)		425 (79.3%)	462 (80.6%)	
Mixed (*n*, %)	90 (10.5%)	45 (10.1%)		65 (12.1%)	89 (15.5%)	

**Table 2 ijms-26-09252-t002:** List of the lead variants from global and sex-stratified analyses.

Gene Variant	Chrom.	Position (GRCh38)	Non-Effect Allele	Effect Allele	Effect Allele Frequency	β	Standard Error	GWAS *p*-Value	Gene Variant– Sex Interaction *p*-Value ^1^
Females and males combined									
rs6664760	1	3,731,704	T	C	0.035	0.857	0.188	5.0 × 10^−6^	1
rs78875161	1	39,627,261	C	T	0.034	−0.800	0.167	1.5 × 10^−6^	1
rs114927020	1	68,967,153	G	A	0.099	−0.518	0.102	4.2 × 10^−6^	1
rs11209392	1	69,122,167	T	G	0.016	−1.194	0.260	4.3 × 10^−6^	1
rs115473801	1	157,613,190	G	A	0.027	−0.891	0.193	3.7 × 10^−6^	1
rs62144353	2	42,755,360	C	A	0.111	0.467	0.100	3.0 × 10^−6^	1
rs12469388	2	177,824,304	G	T	0.030	−0.908	0.183	6.7 × 10^−6^	1
rs1190423	2	232,449,952	A	C	0.341	−0.298	0.064	3.3 × 10^−6^	1
rs62247762	3	73,537,975	C	T	0.024	−0.989	0.204	1.3 × 10^−6^	1
rs6844280	4	31,385,522	C	T	0.487	−0.293	0.061	1.6 × 10^−6^	1
rs116642489	4	152,949,693	C	G	0.065	−0.558	0.122	4.8 × 10^−6^	1
rs113204206	8	1,585,368	C	G	0.105	−0.474	0.099	1.6 × 10^−6^	1
rs11181842	12	43,007,535	C	T	0.161	−0.387	0.081	1.9 × 10^−6^	1
Only females									
rs146309770	4	160,217,229	T	TA	0.035	0.613	0.131	2.7 × 10^−6^	1.6 × 10^−2^
rs555775	9	75,445,184	C	T	0.034	−0.535	0.117	4.6 × 10^−6^	1.0 × 10^−1^
rs915401	9	131,013,384	A	G	0.099	0.419	0.089	2.6 × 10^−6^	9.1 × 10^−5^
rs62070136	17	72,195,002	T	G	0.016	−0.884	0.189	3.1 × 10^−6^	2.1 × 10^−1^
rs11086053	19	17,119,562	A	C	0.027	−0.693	0.148	2.6 × 10^−6^	1
Only males									
rs10888603	1	38,972,945	A	C	0.035	−0.592	0.109	5.2 × 10^−8^	5.5 × 10^−2^
rs2717199	4	111,595,181	A	G	0.034	−0.478	0.094	3.1 × 10^−7^	1.8 × 10^−3^
rs2186409	11	61,127,020	T	G	0.099	−0.496	0.097	2.8 × 10^−7^	7.8 × 10^−4^
rs1274686	19	50,495,701	C	T	0.016	0.832	0.179	3.5 × 10^−6^	1.0 × 10^−5^
rs11700596	21	46,454,141	G	C	0.027	0.522	0.108	1.3 × 10^−6^	1.2 × 10^−1^

^1^ Bonferroni-adjusted *p*-value for gene variant–sex interaction.

## Data Availability

The data that support the findings of this study are available from the corresponding authors upon request.

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
