# Peer review of "Global and Sex-Stratified Genome-Wide Association Study of Long COVID Based on Patient-Driven Symptom Recall"

_ijms, 2025, doi:10.3390/ijms26189252_

Round 1

Reviewer 1 Report

Comments and Suggestions for Authors

In the present study, the authors explored the global and sex-specific genetic variants associated with long COVID of a 1-year cohort  of 2,411 COVID-19 patients with long COVID symptoms.  They have performed global and sex-stratified genome-wide association analyses, including sex-variant interactions,  gene-based analyses, gene-mapping, and gene-set enrichment analyses, and they have identified 23 lead variants from suggestive signals; five variants showed a significant interaction with sex (2 in females, 3 in males) and, among 15 mapped proteins, CD5 and VPS37C, linked to immune function, were significantly associated with long COVID in men, suggesting that persistent immune dysregulation may be involved in the development of precisely defined long COVID. This is a well-organised and well-presented study. The abstract and introduction require minor revisions, the methodology used is adequate, the results are clearly presented, the discussion and conclusions are consistent with the results. The references are appropriate and up-to-date.

Minor comments:

  • Please add the background of the study in the first sentence of the introduction.
  • Please add, cite and discuss the findings from previous studies on genome-wide association analyses of long COVID patients.

Author Response

We sincerely thank the reviewer for their constructive comments, which have helped us to improve the clarity and completeness of our manuscript. Below we provide a point-by-point response.

Comment 1:

Please add the background of the study in the first sentence of the introduction.

Response:

We thank the reviewer for this suggestion. We have now added a general background statement at the beginning of the Introduction to clearly contextualize the study. [Lines 65-67, page 2].

Comment 2:

Please add, cite and discuss the findings from previous studies on genome-wide association analyses of long COVID patients.

Response:

We have now expanded the Discussion to address this and discuss its discrepancy. Lammi et al. reported a genome-wide significant association at the FOXP4 locus when using controls from the general population, which included both SARS-CoV-2–infected and non-infected individuals. However, when their analysis was restricted to infected controls only, the FOXP4 association did not remain statistically significant. Our study used this latter approach by design, as our controls were exclusively SARS-CoV-2–infected individuals without long COVID. This methodological difference, along with our smaller sample size, likely explains why the FOXP4 association was not observed in our results. We now highlight that the choice of control group is crucial in long-COVID GWAS analyses. [Lines 198-208, page 8].

Reviewer 2 Report

Comments and Suggestions for Authors

            The authors analyzed a cohort of patients with symptoms of long-COVID followed at least during a year, and performed GWAS. The study is interesting. Some concerns should be addressed before acceptance

  1. No information is reported on SARS-CoV-2 reinfection cases among this cohort. Reinfection may exacerbate some of the symptoms of an apparent long-COVID. Was this information available? This may be important to include it or at least discuss it.
  2. Why the authors restricted their analysis to sex variants and did not even mention the biological functions of the 13 other genes identified without sex stratification? This should be discussed.
  3. The previous study cited by the authors (Lammi et al., ref 12) identified FOXP4 (gene variant rs9367106). This gene variant was not identified in this study. The authors should explain this difference and discuss these findings.
  4. Line 108: please add ¨The gene variant¨ before its identification.

Author Response

We sincerely thank the reviewer for their constructive comments, which have helped us to improve the clarity and completeness of our manuscript. Below we provide a point-by-point response.

Comment 1:

No information is reported on SARS-CoV-2 reinfection cases among this cohort. Reinfection may exacerbate some of the symptoms of an apparent long-COVID. Was this information available? This may be important to include it or at least discuss it.

Response:

We appreciate this important observation. In our cohort, information on reinfections was not systematically collected during follow-up, and therefore we cannot exclude that some participants may have experienced a reinfection. We have now acknowledged this as a limitation of our study and included a note in the Discussion to highlight that reinfections may contribute to persistent or recurrent symptoms and could partially influence long-COVID phenotypes. Future studies should systematically evaluate reinfection events to better clarify their role. [Lines 223-226, page 8].

Comment 2:

Why the authors restricted their analysis to sex variants and did not even mention the biological functions of the 13 other genes identified without sex stratification? This should be discussed.

Response:

We agree with the reviewer, but our main objective was to investigate sex-specific associations with long COVID, given the well-established differences in prevalence and clinical presentation between female and male. For this reason, we focused the functional annotation and discussion on the genes identified in the sex-stratified analyses, as these findings provide more specific mechanistic hypotheses and pathways that may underlie sex differences in long-COVID susceptibility. While the global analysis identified 13 additional loci, we decided not to expand on these results in detail to keep the discussion aligned with our primary aim.

Comment 3:

The previous study cited by the authors (Lammi et al., ref 12) identified FOXP4 (gene variant rs9367106). This gene variant was not identified in this study. The authors should explain this difference and discuss these findings.

Response:

We have now expanded the Discussion to address this discrepancy. Lammi et al. reported a genome-wide significant association at the FOXP4 locus when using controls from the general population, which included both SARS-CoV-2–infected and non-infected individuals. However, when their analysis was restricted to infected controls only, the FOXP4 association did not remain statistically significant. Our study used this latter approach by design, as our controls were exclusively SARS-CoV-2–infected individuals without long COVID. This methodological difference, along with our smaller sample size, likely explains why the FOXP4 association was not observed in our results. We now highlight that the choice of control group is crucial in long-COVID GWAS analyses. [Lines 198-208, page 8].

Comment 4:

Line 108: please add "The gene variant" before its identification.

Response:

We have implemented this correction as requested. The sentence now reads: “The gene variant rs10888603, …”. [Line 111, page 4].

Round 2

Reviewer 2 Report

Comments and Suggestions for Authors

         The authors addressed satisfactorely the concerns.